

# Ground state analysis of the spin-1/2 XX chain model with anisotropic three-spin interaction

Nima Mahboob[1], Saeed Mahdavifar[1][*] and Fatemeh Khastehdel Fumani[2]

**1** Department of Physics, University of Guilan, 41335-1914, Rasht, Iran
**2** Department of Basic Sciences, Lan.C., Islamic Azad University, Langarud, Iran

[*] smahdavifar@gmail.com

## Abstract

In this paper, we investigate the ground state properties of the spin-1/2 XX chain model with anisotropic three-spin interaction using the fermionization technique. By exactly diagonalizing the Hamiltonian, we analyze the dispersion relation, ground state energy, and order parameters. Our findings identify two gapless phases, distinguished by a Lifshitz transition line: one exhibiting long-range chiral correlations and the other characterized by chiral-nematic correlations. These correlations do not correspond to conventional symmetry-breaking local order parameters typically associated with gapped one-dimensional phases but instead signify emergent quantum coherence in the gapless regime. Further, we study the ground state phase diagram through the concurrence and quantum discord between nearest-neighbor spins, finding that these measures are maximized at the critical line, with an additional entangled region observed. Finally, we examine the spin squeezing parameter and entanglement entropy, demonstrating that the ground state is squeezed throughout and becomes extremely squeezed at the critical line. Notably, in the gapless chiral-nematic phase, the Heisenberg limit is achieved. By dividing the system into two equal parts, we observe significant entanglement in the gapless chiral phase. The central charge calculation confirms the critical nature of the gapless chiral-nematic phase, while the entanglement entropy follows volume-law scaling in the gapless chiral phase.

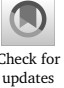

# 1  Introduction

The quantum ground state represents the lowest energy configuration of a system, where all particles occupy minimum energy levels, even at absolute zero temperature. Understanding its phase is crucial for exploring quantum phenomena, especially in low-dimensional quantum magnets [1, 2]. A ground state phase diagram maps the different phases as parameters like pressure or magnetic field vary, revealing quantum phase transitions driven by fluctuations [3, 4]. This diagram is essential in material science for designing superconductors and magnetic materials, in quantum computing for optimizing algorithms and qubit behavior, and in condensed matter physics for studying superconductivity, magnetism, and topological phases.

Quantum phase transitions can involve changes in energy gaps. In a gapped-gapped transition, two distinct phases maintain an energy gap, as seen in spin-1/2 chains like the Haldane phase, which has unique topological properties [5]. In contrast, a gapped-gapless transition occurs when a system shifts from a gapped state to a gapless one, such as the transition to a Luttinger liquid phase in the XXZ spin-1/2 chain by tuning anisotropy [6–8].

A gapless-gapless phase transition refers to a transition between two different gapless phases in a quantum system. In these transitions, the system remains gapless before and after the transition, but the nature of the gapless excitations changes. In spin-1/2 chain systems the three-spin interaction (TSI) can induce a gapless-gapless quantum phase transition. This is a type of cluster interaction, where a group of spins interact collectively. In particular, TSI is when three spins are coupled by a term that depends on the product of their spin components. TSI can cause frustration, entanglement, and quantum phase transitions in spin systems. Two different type of TSI are well known. The Hamiltonian of the first model is given by

$$\mathcal{H} = J^{'} \sum_{n=1}^{N} \left[ \left( S_n^x S_{n+2}^x + S_n^y S_{n+2}^y \right) S_{n+1}^z \right], \tag{1}$$

where $S_n$ is the spin operator at the $n$-th site and $J^{'}$ denotes the strength of the TSI. The impact of this type of TSI on the spin-1/2 Heisenberg chain model has been extensively studied [9–27]. Notably, when this interaction is added to the spin-1/2 XX chain model, it induces a gapless-gapless quantum phase transition [10].

The Hamiltonian of the second model is expressed as

$$\mathcal{H} = J^{'} \sum_{n=1}^{N} \left( S_n^x S_{n+2}^y - S_n^y S_{n+2}^x \right) S_{n+1}^z. \tag{2}$$

The effects of this type of TSI on the spin-1/2 Heisenberg chain model are also well-documented [11, 28–34]. Specifically, when this interaction is introduced to the spin-1/2 XX chain model, a gapless-gapless quantum phase transition is observed [11].

Gapless-gapless quantum phase transitions in spin-1/2 chains are key to condensed matter physics and quantum information science. They provide insights into quantum criticality,

exotic phases like spin liquids, and material behavior at critical points. In quantum technology, gapless systems exhibit unique entanglement properties, aid quantum algorithm development, and contribute to fault-tolerant quantum computation. They also help design quantum materials for sensors and communication while serving as models for testing new quantum technologies.

Recently, there has been significant interest in studying the effects induced by anisotropic three-spin interactions (ATSI). Specially studies are focussed on the anisotropic form of the first model which is defined as

$$\mathcal{H} = J' \sum_{n=1}^{N} \left[ \left( S_n^x S_{n+2}^x + \beta S_n^y S_{n+2}^y \right) S_{n+1}^z \right], \tag{3}$$

where $\beta$ is the anisotropy parameter. Recent studies show that ATSI can induce gapped topological phases in spin-1/2 XY and XX chains [18, 35–39]. Extended quantum Ising models are characterized using winding numbers in a two-dimensional space [18]. Research on entanglement and quantum discord in XY models highlights how anisotropy affects correlations [35], while studies on decoherence examine topological state relaxation [36]. Additionally, long-range quantum resources in XX chains help identify topological phases, with implications for quantum information processing [37].

In this study, we investigate the ground state of the spin-1/2 XX chain model with the second type of ATSI, described by the Hamiltonian:

$$\mathcal{H} = J' \sum_{n=1}^{N} \left[ \left( S_n^x S_{n+2}^y + \beta S_n^y S_{n+2}^x \right) S_{n+1}^z \right]. \tag{4}$$

Our goal is to provide a comprehensive depiction of the ground state phase diagram using tools from both condensed matter physics, quantum information science and quantum measurements. Using the fermionization technique, the exact ground state is obtained. Initially, by analyzing the dispersion relation, ground state energy, and cluster order parameters, two gapless phases separated by a Lifshitz transition line are identified: a phase with chiral long-range ordering, and a phase with chiral-nematic long-range ordering. Next, by examining the concurrence and quantum discord (QD) between nearest-neighbor spins, the ground state phase diagram is studied, revealing that these functions are maximized on the critical line and an entangled region is observed. Finally, the ground state phase diagram is analyzed using the spin squeezing parameter and entanglement entropy (EE). Results show that the ground state is squeezed throughout and becomes extremely squeezed at the critical line. Surprisingly, in the gapless chiral-nematic phase, the Heisenberg limit is achieved. Dividing the system into two equal parts reveals high entanglement in the gapless chiral phase. The central charge is calculated, showing that the gapless chiral-nematic phase is critical based on EE scaling, and EE follows volume-law scaling in the gapless chiral phase.

The structure of the paper is as follows: In the next section, we introduce the model and determine the ground state using the fermionization technique. Section III presents the results. Finally, Section IV concludes with our findings.

## 2 Model

The Hamiltonian of the 1D spin-1/2 XX model with ATSI is denoted as

$$\mathcal{H} = J \sum_{n=1}^{N} \left[ S_n^x S_{n+1}^x + S_n^y S_{n+1}^y \right] + J' \sum_{n=1}^{N} \left[ \left( S_n^x S_{n+2}^y + \beta S_n^y S_{n+2}^x \right) S_{n+1}^z \right], \tag{5}$$

where symbol $J > 0$ represents the antiferromagnetic exchange coupling. The variable $N$ denotes the size of the system, which corresponds to the number of spins. We are assuming the periodic boundary condition, denoted as $S_{n+N}^{\mu} = S_n^{\mu}$, where $\mu$ might take the directions of $x$, $y$, or $z$. Assuming $\alpha = \frac{J'}{J}$ is considered valid and does not affect the generality of the situation.

## 2.1 Symmetry arguments

Consider a unitary transformation defined as $U_z = \prod_{n=1}^{N} e^{i\pi n S_n^z}$, which results in the following transformations:

$$S_n^{x,y} \longrightarrow (-1)^n S_n^{x,y}, \qquad S_n^z \longrightarrow S_n^z. \tag{6}$$

When $N$ is even, the Hamiltonian satisfies the symmetry relation:

$$U_z \mathcal{H}(J, J', \beta) U_z^{\dagger} = \mathcal{H}(-J, J', \beta), \tag{7}$$

indicating that the system's features for $J < 0$ can be reconstructed using this transformation. Additionally, by time reversal transformation, $\tau$, we obtain:

$$\tau \mathcal{H}(J, J', \beta) \tau^{\dagger} = \mathcal{H}(J, -J', \beta), \tag{8}$$

allowing the properties of the system for $J' < 0$ to be derived from the ground state phase diagram for $J' > 0$. Finally, under the combined symmetry operator $W = \tau U_z$, the Hamiltonian transforms as:

$$W \mathcal{H}(J, J', \beta) W^{\dagger} = -\mathcal{H}(J, J', \beta), \tag{9}$$

which implies that the energy spectrum of the Hamiltonian exhibits inversion symmetry around zero energy, reflecting a particle-hole-like symmetry. This further suggests that zero-energy states must appear in pairs if present in the spectrum.

Therefore the ATSI disrupts several symmetries in the pure 1D spin-1/2 XX model, specifically:

- the $U(1)$ symmetry,

- time reversal symmetry,

- the parity symmetry.

In the rest of this paper we show that the breaking of $U(1)$ symmetry, time reversal symmetry, and parity symmetry in the spin-1/2 XX chain model with ATSI significantly influences the ground state properties and phase behaviors. They leads to the emergence of chiral and chiral-nematic long-range orderings, enhancing quantum correlations such as concurrence and quantum discord. Time reversal symmetry breaking is directly related to the presence of chiral order and influences the critical behavior, enhancing quantum correlations at the critical line. Parity symmetry breaking contributes to the unique long-range ordering in the chiral-nematic phase and affects entanglement properties and spin squeezing, leading to enhanced quantum correlations and squeezing.

Overall, the interplay between these symmetry breakings and the ATSI results in a rich tapestry of ground state properties, including distinct gapless phases, Lifshitz critical line, and regions of enhanced quantum correlations. These findings provide valuable insights into quantum phase transitions and the role of multi-spin interactions in low-dimensional quantum systems, with potential implications for the development of quantum technologies.

## 2.2  Diagonalization

The Hamiltonian (Eq. (5)) can be exactly diagonalized using the fermionization approach. By applying the Jordan-Wigner transformation as follows:

$$S_n^+ = a_n^\dagger e^{i\pi\left(\sum_{l<n} a_l^\dagger a_l\right)}, \qquad S_n^z = a_n^\dagger a_n - \frac{1}{2}, \tag{10}$$

where $a_n^\dagger$ and $a_n$ are the fermionic operators, the fermionized noninteracting form of the Hamiltonian is obtained as:

$$\begin{aligned}
\mathcal{H} = \frac{1}{2}\sum_{n=1}^{N}\left(a_n^\dagger a_{n+1} + a_{n+1}^\dagger a_n\right) + \frac{i}{8}\alpha(\beta-1)\sum_{n=1}^{N}\left(a_n^\dagger a_{n+2} - a_{n+2}^\dagger a_n\right) \\
+ \frac{i}{8}\alpha(\beta+1)\sum_{n=1}^{N}\left(a_n^\dagger a_{n+2}^\dagger - a_{n+2}a_n\right).
\end{aligned} \tag{11}$$

By performing a Fourier transformation as $a_n = \frac{1}{\sqrt{N}}\sum_k e^{-ikn}a_k$, and also a Bogoliubov transformation:

$$a_k = \cos(\theta_k)\gamma_k - \sin(\theta_k)\gamma_{-k}^\dagger, \tag{12}$$

the diagonalized Hamiltonian is obtained as:

$$\mathcal{H} = \sum_{k=-\pi}^{\pi} \varepsilon(k)\left(\gamma_k^\dagger \gamma_k - \frac{1}{2}\right). \tag{13}$$

The single-particle dispersion relation or the energy spectrum is given by:

$$\varepsilon(k) = \sqrt{\mathcal{A}_k^2 + \mathcal{C}_k^2} + \mathcal{B}_k, \tag{14}$$

with:

$$\mathcal{A}_k = \cos(k), \quad \mathcal{B}_k = \frac{1}{4}\alpha(\beta-1)\sin(2k), \quad \mathcal{C}_k = \frac{1}{4}\alpha(\beta+1)\sin(2k), \quad \tan(2\theta_k) = \frac{\mathcal{C}_k}{\mathcal{A}_k}. \tag{15}$$

The summation in Eq. (13) is carried out over values of $k = 2\pi m/N$, where $m$ takes the range $0, \pm 1, \ldots, \pm\frac{1}{2}(N-1)$ for odd $N$, and $m = 0, \pm 1, \ldots, \pm(\frac{1}{2}N-1), \frac{1}{2}N$ for even $N$, assuming periodic boundary conditions have been imposed on the Jordan-Wigner fermions.

In the thermodynamic limit, $N \to \infty$, the ground state of the system corresponds to the configuration where all the states with $\varepsilon(k) \leq 0$ are filled and those with $\varepsilon(k) > 0$ are empty.

For $\beta > 0$, there are only two Fermi points located at

$$K_F = \pm\frac{\pi}{2},$$

and the ground state is the vacuum state, $|GS\rangle = |0\rangle$. In the region where $\beta < 0$, if $-\beta \leq \frac{1}{\alpha^2}$, two additional Fermi points emerge at

$$K_F^+ = \arcsin\left(\frac{1}{\alpha\sqrt{-\beta}}\right), \qquad K_F^- = -\pi + \arcsin\left(\frac{1}{\alpha\sqrt{-\beta}}\right). \tag{16}$$

In this case, the ground state is characterized by all states being filled within region $\Lambda$, which is identified by the conditions $-\pi < K < K_F^-$ and $K_F^+ < K < \pi$ ($|GS\rangle = \prod_{k\in\Lambda}\gamma_k^\dagger|0\rangle$). Consequently, a quantum phase transition occurs at

$$\beta_c = -\frac{1}{\alpha^2}. \tag{17}$$

This transition is between two gapless phases: one with two Fermi points and the other with four Fermi points.

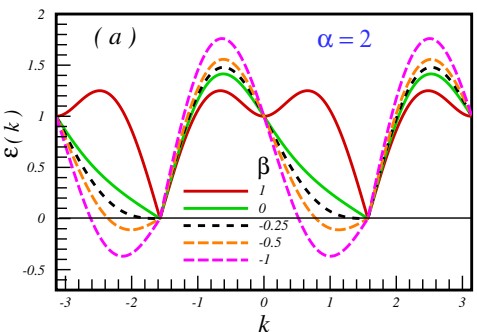
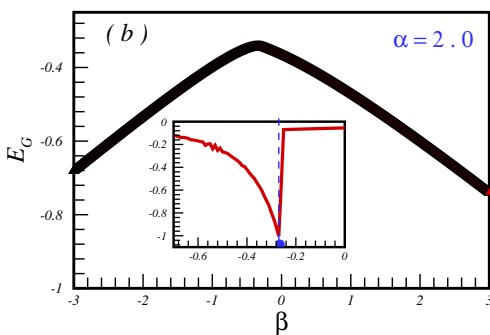

Figure 1: (color online) (a) The spectra for various values of $\beta$ and $\alpha = 2.0$ are shown for a chain system with $N = 1000$ particles. (b) The ground state energy density as a function of $\beta$ at $\alpha = 1.0$ is presented, with the second derivative of the ground state energy density plotted in the inset.

## 3 Results

### 3.1 Phase transition in the ground state

We begin our study by examining the dispersion relation, a crucial tool for investigating the microscopic properties of quantum systems and understanding quantum phase transitions. The dispersion relation describes how the energy of quasi-particles varies with their momentum, essentially mapping the distribution of energy levels as a function of the wave vector. It offers insights into the nature of elementary excitations within the system. For instance, in a gapped phase, the dispersion relation reveals a minimum energy gap, whereas in a gapless phase, it reaches zero energy at certain points. Near quantum phase transitions, the dispersion relation often exhibits critical behavior, such as the appearance of new low-energy excitations or alterations in the spectrum, signaling a transition from one phase to another.

In Fig. 1 (a), we have plotted $\varepsilon(k)$ as a function of $k$ for a chain system of size $N = 1000$, which is sufficiently large to be regarded as the thermodynamic limit. We consider $\alpha = 2.0$ with different values of $\beta = 1, 0, -0.25, -0.5, -1.0$. The critical point in this case is $\beta_c = -0.25$. For $\beta > \beta_c$, the system exhibits two Fermi points, indicating a particular topological structure of the Fermi surface. However, when $\beta < \beta_c$, a significant change in the topology occurs, leading to the emergence of four Fermi points. Despite this structural transformation, both regions remain gapless, highlighting the presence of a Lifshitz transition in our model.

A gapless-gapless Lifshitz transition is a type of quantum phase transition where the system maintains a gapless spectrum before and after the transition, yet the topology of its low-energy excitations undergoes a fundamental shift. In many cases, such transitions manifest through changes in the number or configuration of Fermi points, altering the nature of quasiparticle dynamics. The behavior observed in this figure provides clear evidence of such a transition within our model, emphasizing the crucial role played by topological modifications in defining quantum critical behavior.

The ground-state energy density of the system is expressed as:

$$E_G = \frac{1}{2N} \sum_{k \in \Lambda} \varepsilon(k) - \frac{1}{2N} \sum_{k \notin \Lambda} \varepsilon(k), \tag{18}$$

where $\Lambda$ represents the region where $\varepsilon(k) < 0$. In Fig. 1 (b), we have plotted the ground-state energy density of the system as a function of $\beta$ for $\alpha = 2.0$. The inset shows the second derivative of the ground-state energy density as a function of $\beta$. It is evident that there is a

discontinuity in the second derivative of the ground-state energy density, $\partial^2 E_G/\partial \beta^2$, precisely at the critical point $\beta_c = -0.25$, indicating a continues quantum phase transition.

In the realm of quantum many-body systems, response functions and specially order parameters are crucial indicators that delineate distinct ground state domains by manifesting specific types of order. These parameters are zero in disordered phases, reflecting symmetry, and become finite in ordered phases, indicating broken symmetry. The significance of order parameters extends beyond mere classification; they are instrumental in understanding the nature of quantum phase transitions. Order parameters provide valuable insights into the character and classification of these transitions, which can either unfold gradually as continuous transitions or occur abruptly as discontinuous transitions. By analyzing order parameters, we gain a deeper understanding of the fundamental mechanisms governing the behavior of quantum many-body systems and the intricate nature of their phase transitions.

Cluster order parameters, constructed through the interaction of three or more spin operators, are widely recognized as effective indicators of exotic quantum phases. Although they may not conform to the traditional definition of long-range order parameters, cluster order parameters offer a robust means of capturing localized spin correlations and uncovering emergent phenomena within quantum systems. By establishing their relationship to symmetry-breaking mechanisms and their role in distinguishing between different phases, these parameters can be confidently validated as order parameters within the framework of specific spin-1/2 systems. It is important to note that this model does not induce any magnetization. Therefore, to thoroughly understand the nature of the various gapless phases within our model, we have calculated the cluster order parameters as defined:

$$
\begin{aligned}
Cl^- &= \frac{1}{N} \sum_{n=1}^{N} \left\langle \left( S_n^x S_{n+2}^y - S_n^y S_{n+2}^x \right) S_{n+1}^z \right\rangle \\
&= -\frac{1}{2N} \sum_{k \in \Lambda} \sin(2k) \\
&= \begin{cases} -\frac{1}{2\pi}\left(1 + \frac{1}{\alpha^2 \beta}\right), & \text{for } \beta < 0, \\ 0, & \text{for } \beta > 0, \end{cases} \\
Cl^+ &= \frac{1}{N} \sum_{n=1}^{N} \left\langle \left( S_n^x S_{n+2}^y + S_n^y S_{n+2}^x \right) S_{n+1}^z \right\rangle \\
&= \frac{1}{4N} \sum_{k \in \Lambda} \sin(2k) \sin(2\theta_k) - \frac{1}{4N} \sum_{k \notin \Lambda} \sin(2k) \sin(2\theta_k).
\end{aligned}
\tag{19}
$$

In terms of symmetry properties, both parameters violate time-reversal symmetry and partially break spin rotational symmetry. However, their key distinction lies in parity: $Cl^-$ disrupts parity symmetry, whereas $Cl^+$ maintains it.

On the other hand, these cluster order parameters can be understood as connected correlators, establishing relationships between spin components at different lattice sites as follows:

$$
S_n^x S_{n+2}^y S_{n+1}^z \propto \left( S_n^x S_{n+1}^y \right)\left( S_{n+1}^x S_{n+2}^y \right), \qquad S_n^y S_{n+2}^x S_{n+1}^z \propto \left( S_n^y S_{n+1}^x \right)\left( S_{n+1}^y S_{n+2}^x \right).
\tag{20}
$$

These expressions highlight how correlations extend beyond nearest-neighbor interactions, contributing to the underlying order present in the system. The cluster order parameter $Cl^-$ is also referred to as the scalar chirality parameter [40]. A non-zero value of $Cl^-$ indicates the presence of chiral ordering. The chiral order parameter serves as a measure of the helical spin alignment along a chain. In spin-1/2 chain models, chiral order can originate from various sources, including competing interactions. These chiral fluctuations play a crucial role in inducing quantum phase transitions between different chiral states or between chiral and

non-chiral states. Importantly, a chiral phase is marked by the breaking of time-reversal and parity symmetries without the emergence of magnetic order.

The cluster order parameter $Cl^+$ appears to have a notable connection to spin nematicity, as suggested in references [41–44]. Spin nematicity is a phenomenon observed in quantum spin systems, where the spins exhibit alignment along two distinct axes in the chain, rather than a single axis typical of standard spin arrangements. This phase, known as the quantum spin-nematic phase, represents an exotic state of matter and is characterized by the presence of spin quadrupolar order, where spins show orientational correlations along two perpendicular directions instead of being aligned along a singular direction. The spin-nematic phase thus quantifies the degree of quadrupolar order in the system.

A key aspect of $Cl^+$ is its deviation from the conventional spin-nematic parameter. While spin nematicity traditionally focuses on quadrupolar correlations and maintains time-reversal symmetry, $Cl^+$ incorporates an additional term that introduces chiral-like correlations. This extra term reflects the presence of mixed spin components interacting along different directions, suggesting a more complex structure. $Cl^+$ does not exclusively describe quadrupolar correlations. The combination of mixed spin components implies that $Cl^+$ exhibits chirality, in addition to nematicity, breaking the pure time-reversal symmetry associated with conventional spin-nematic states. As a result, $Cl^+$ represents a hybrid order parameter that merges elements of chirality (from the directional coupling of spin components) with nematicity (from quadrupolar-like orientational correlations).

Due to its unique properties that integrate features of both chirality and nematicity, $Cl^+$ is termed a chiral-nematic order parameter. This classification emphasizes the dual nature of $Cl^+$, which accounts for the unconventional correlations that arise in certain spin systems. By bridging chiral and nematic characteristics, the chiral-nematic order parameter provides a deeper understanding of the emergent phenomena in complex spin chain systems. This expanded view highlights the importance of $Cl^+$ as a more generalized and versatile descriptor of the order in spin chains, especially in cases where conventional nematic descriptions fall short.

In the case of the pure ATSI model with $J = 0$, the cluster order parameters exhibit distinct behavior based on the sign of $\beta$ as

$$Cl^- = \begin{cases} -\frac{1}{2\pi}, & \text{for } \beta < 0, \\ 0, & \text{for } \beta > 0, \end{cases} \qquad Cl^+ = \begin{cases} 0, & \text{for } \beta < 0, \\ -\frac{1}{2\pi}, & \text{for } \beta > 0. \end{cases}$$

When $\beta < 0$, the $Cl^-$ takes on a finite value of $-\frac{1}{2\pi}$, $Cl^+$ remains at zero, indicating a preference for chiral ordering. This phase is characterized by a spontaneous breaking of parity symmetry, where local spin orientations adopt non-trivial rotational structures. As a result, the system favors the chiral phase, leading to strong correlations between spins. Conversely, when $\beta > 0$, the roles of the order parameters reverse $Cl^-$ vanishes while $Cl^+$ acquires the finite value $-\frac{1}{2\pi}$, marking the chiral-nematic ordering.

At the specific value $\beta = -1$, the ground-state phase diagram of the model has been previously established [11]. In our investigation, at $\beta = -1$, we find that $\sin(K_F^+) = \frac{1}{\alpha}$, and the function $\mathcal{C}_k$ vanishes, leading to the condition $\tan(2\theta_k) = 0$. Substituting these values into the expressions for the cluster order parameters, we obtain:

$$Cl^- = \begin{cases} 0, & \text{for } \alpha \leq 1, \\ -\frac{1}{2\pi}\left(1 - \frac{1}{\alpha^2}\right), & \text{for } \alpha > 1. \end{cases}$$

Additionally, the order parameter $Cl^+$ remains zero across all values of $\alpha$, which is in complete agreement with previous studies. This result further confirms the consistency of the phase structure in this regime and validates the theoretical framework used to analyze quantum correlations in the system.

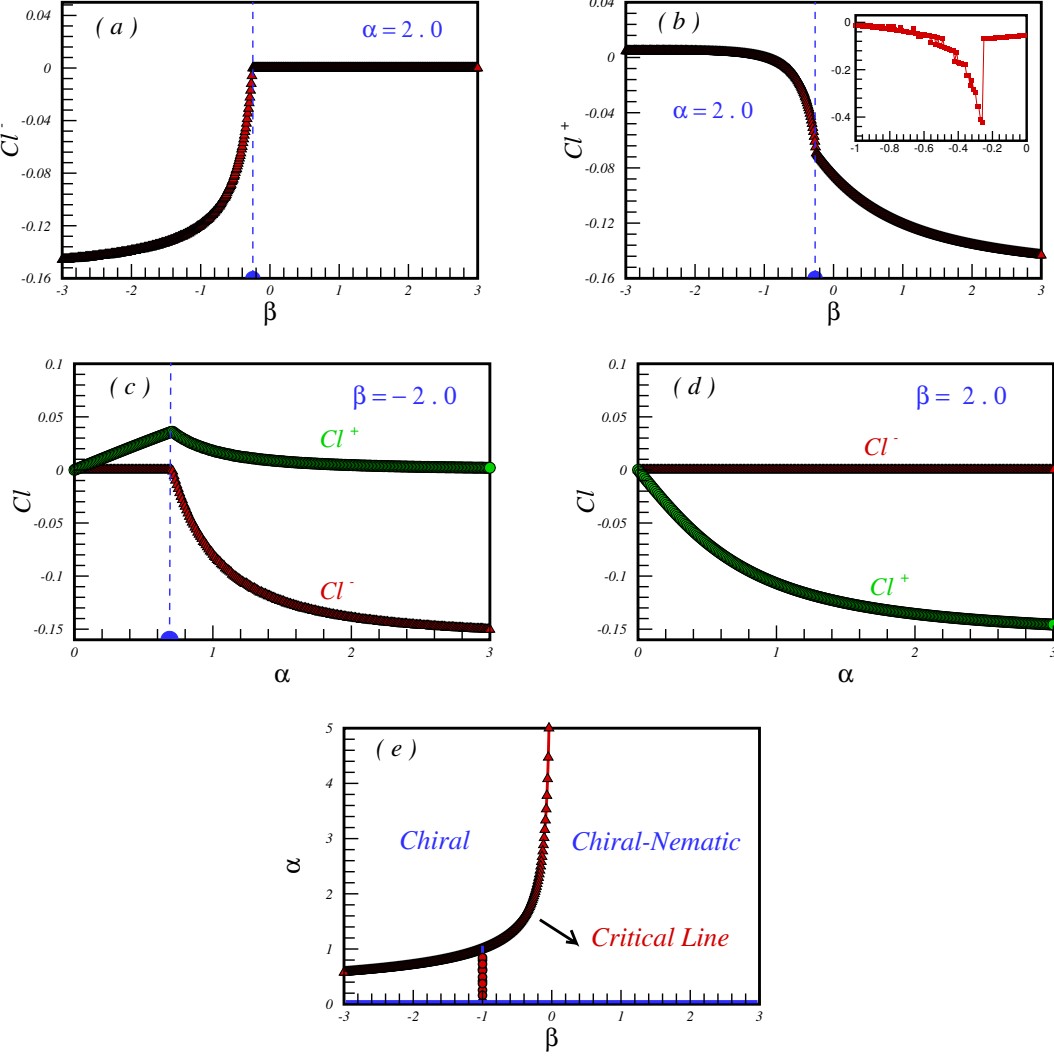

Figure 2: (color online) The cluster order parameters (a) $Cl^-$ and (b) $Cl^+$ as functions of $\beta$ at $\alpha = 1.0$ for a chain system with $N = 1000$ particles. In the inset of (b), results for the first derivative of $Cl^+$ with respect to $\beta$ are shown. (c) and (d) show the cluster order parameters with respect to $\alpha$ for $\beta = -2$ and $\beta = 2$, respectively. (e) The ground state phase diagram in the $\beta$-$\alpha$ plane. At $\alpha = 0$, where the system reduces to the pure spin-1/2 XX chain, the ground state resides in the Luttinger liquid phase. Similarly, when $\beta = -1$ and $\alpha < 1$, the ground state remains in the Luttinger liquid phase. Along these lines, no evidence of cluster ordering is detected.

The results for the cluster order parameters are depicted in Fig. 2 (a) and (b) for a chain size of $N = 1000$ and $\alpha = 2.0$. As shown in Fig. 2 (a), $Cl^-$ act as order parameter. In the region where $\beta < \beta_c = -0.25$, the ground state exhibits chiral phase characteristics with a non-zero chirality value. Conversely, as the system transitions to the region where $\beta > \beta_c = -0.25$, the chirality vanishes. Thus, chiral ordering is present in the region $\beta < \beta_c$, and the spectrum remains gapless.

On the other hand, Fig. 2 (b) illustrates the behavior of $Cl^+$ across different regions of $\beta$. In the region where $\beta < \beta_c = -0.25$, $Cl^+$ exhibits an extremely small domain, almost negligible, which is primarily attributed to the effects of quantum fluctuations. Its value transitions very smoothly as $\beta$ varies, reflecting the subtle interplay of quantum phenomena in this regime.

Conversely, in the gapless region where $\beta > \beta_c = -0.25$, a stark contrast is observed. The value of $Cl^+$ changes substantially with $\beta$, revealing characteristics that are indicative of the chiral-nematic phase. This distinct behavior underscores the emergence of novel ordering properties in this phase. Moreover, the inset provides further clarity by examining the first derivative of $Cl^+$. It identifies a quantum critical point at $\beta = \beta_c = -0.25$, marking a pivotal transition in the system's ground state. Beyond this critical point, in the region $\beta > \beta_c$, chiral-nematic ordering is clearly present, signifying the unique structural organization of the phase. Notably, throughout this region, the system maintains its gapless nature, reinforcing the significance of quantum fluctuations and critical behavior in shaping the properties of the chiral-nematic phase.

The variation of cluster order parameters with respect to $\alpha$ is presented in Fig. 2, where panel (c) corresponds to $\beta = -2.0$ and panel (d) to $\beta = 2.0$. It is evident that cluster ordering does not occur in the absence of cluster interaction ($\alpha = 0$). The results explicitly indicate that for $\beta = -2.0$, a quantum phase transition takes place at $\alpha_c = \frac{1}{\sqrt{2}}$, whereas no such transition is observed for $\beta = 2.0$. Moreover, chiral ordering is only observed when $\alpha > \alpha_c$.

Furthermore, as depicted in Fig. 2 (d), an increase in $\alpha$ from zero immediately triggers chiral-nematic ordering, causing $Cl^+$ to rise as $\alpha$ increases. In the region where $\beta = -2.0 < 0$ and $\alpha < \alpha_c = \frac{1}{\sqrt{2}}$ (Fig. 2 (c)), $Cl^+$ continues to increase, reaching a maximum at $\alpha_c = \frac{1}{\sqrt{2}}$. However, upon entering the second region ($\alpha > \frac{1}{\sqrt{2}}$), $Cl^+$ declines significantly, assuming very small values due to the influence of quantum fluctuations.

Finally, the ground state phase diagram is presented in Fig. 2 (e), illustrating the intricate interplay of phases within the model. Notably, along the line $\alpha = 0$, the model simplifies to the spin-1/2 XX chain, where the ground state exhibits the gapless Luttinger liquid phase, a hallmark of quantum critical behavior. Additionally, for $\beta = -1$, it has been demonstrated that the ground state remains within the Luttinger liquid phase for values of $\alpha$ up to 1. At this point, the system undergoes a remarkable phase transition, giving rise to the chiral phase, which reflects a profound structural transformation. Moreover, the Lifshitz critical line, defined by $\beta = -\frac{1}{\alpha^2}$, emerges as a boundary separating two distinct gapless regions for $\alpha \neq 0$. In the regime where $\beta < \beta_c$, the system is characterized by the presence of long-range chiral ordering, signifying a highly correlated quantum state. Conversely, for $\beta > \beta_c$, the model transitions into another gapless phase, distinguished by long-range chiral-nematic ordering. This region represents a fascinating interplay of quantum fluctuations and symmetry-breaking mechanisms that drive the emergence of unique ordering phenomena. The insights gleaned from the phase diagram highlight the richness and diversity of quantum phases within the model, emphasizing its utility in exploring critical transitions and ordered states in quantum systems.

## 3.2 Concurrence and quantum discord

Quantum correlations are the non-classical correlations that arise between parts of a quantum system due to the principles of quantum mechanics [45–49]. These correlations are fundamentally different from classical correlations and can exhibit unique properties such as entanglement and quantum discord (QD).

When particles become entangled, their quantum states are interdependent, meaning the state of one particle cannot be described independently of the state of the other. This leads to strong correlations between the particles, even when they are separated by large distances.

A widely used method to quantify entanglement is concurrence, applicable to both pure and mixed states of two qubits [50]. Concurrence ranges from zero for separable states to one for maximally entangled states. For two arbitrary spins at positions $i$ and $j$, the two-site

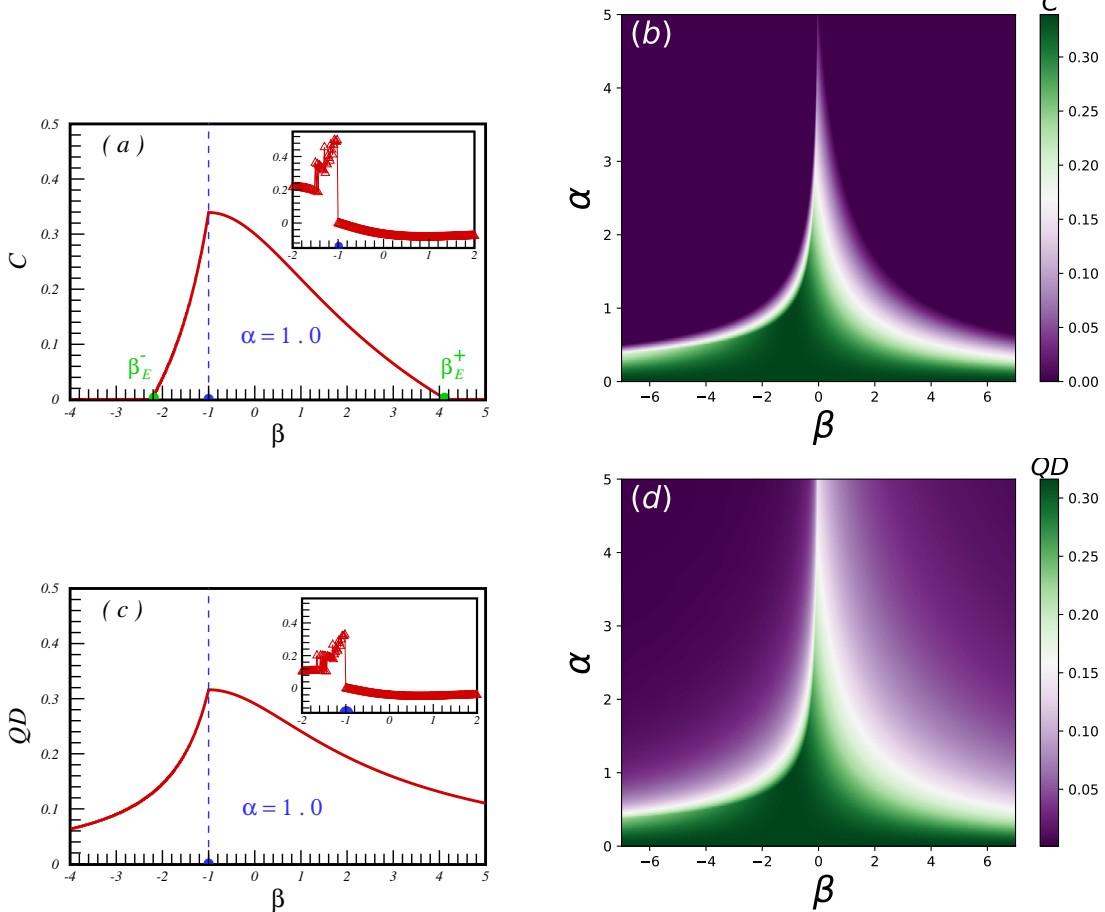

Figure 3: (color online) (a) The concurrence as a function of $\beta$ at $\alpha = 1.0$ for a chain system with $N = 1000$ particles. The inset shows the first derivative of concurrence with respect to $\beta$. (b) The density plot of concurrence in the $\beta$-$\alpha$ plane. (c) The QD as a function of $\beta$ at $\alpha = 1.0$ for a chain system with $N = 1000$ particles. The inset shows the first derivative of QD with respect to $\beta$. (d) The density plot of QD in the $\beta$-$\alpha$ plane.

reduced density matrix typically takes the form

$$\rho_{ij} = \frac{1}{4} + \sum_{\mu}\left(\langle S_i^{\mu}\rangle S_i^{\mu} + \langle S_j^{\mu}\rangle S_j^{\mu}\right) + \sum_{\mu,\nu}\langle S_i^{\mu}S_j^{\nu}\rangle S_i^{\mu}S_j^{\nu}, \tag{21}$$

where $\mu, \nu = x, y, z$. The concurrence between two spin-1/2 particles at sites $i$ and $j$ can be derived from the corresponding reduced density matrix $\rho_{ij}$. The reduced density matrix in the standard basis $(|\uparrow\uparrow\rangle, |\uparrow\downarrow\rangle, |\downarrow\uparrow\rangle, |\downarrow\downarrow\rangle)$ is expressed as

$$\rho_{i,j} = \begin{pmatrix} \langle p_i^{\uparrow}p_j^{\uparrow}\rangle & \langle p_i^{\uparrow}S_j^{-}\rangle & \langle S_i^{-}p_j^{\uparrow}\rangle & \langle S_i^{-}S_j^{-}\rangle \\ \langle p_i^{\uparrow}S_j^{+}\rangle & \langle p_i^{\uparrow}p_j^{\downarrow}\rangle & \langle S_i^{-}S_j^{+}\rangle & \langle S_i^{-}p_j^{\downarrow}\rangle \\ \langle S_i^{+}p_j^{\uparrow}\rangle & \langle S_i^{+}S_j^{-}\rangle & \langle p_i^{\downarrow}p_j^{\uparrow}\rangle & \langle p_i^{\downarrow}S_j^{-}\rangle \\ \langle S_i^{+}S_j^{+}\rangle & \langle S_i^{+}p_j^{\downarrow}\rangle & \langle p_i^{\downarrow}S_j^{+}\rangle & \langle p_i^{\downarrow}p_j^{\downarrow}\rangle \end{pmatrix}, \tag{22}$$

where the brackets denote the physical state average, and $p^{\uparrow} = \frac{1}{2} + S^z$, $p^{\downarrow} = \frac{1}{2} - S^z$, $S^{\pm} = S^x \pm iS^y$. The concurrence between two spins is given by $C = \max(0, \lambda_1 - \lambda_2 - \lambda_3 - \lambda_4)$, where $\lambda_i$ is the square root of the eigenvalue of $R = \rho_{ij}\tilde{\rho}_{ij}$ and $\tilde{\rho}_{ij} = (\sigma_i^y \otimes \sigma_j^y)\rho_{ij}^{\star}(\sigma_i^y \otimes \sigma_j^y)$.

Due to the symmetry of the Hamiltonian, most off-diagonal elements of the reduced density matrix $\rho_{ij}$ will be zero. First, translation invariance requires that the density matrix satisfies $\rho_{ij} = \rho_{i,i+r}$ for any position $i$. Additionally, our model is invariant under $\pi$-rotation around the $z$ direction. Following these symmetry properties, the density matrix must be symmetrical, and only certain elements of the reduced density matrix become non-zero,

$$\rho_{ij} = \begin{pmatrix} X_{i,j}^+ & 0 & 0 & F_{i,j}^* \\ 0 & Y_{i,j}^+ & Z_{i,j}^* & 0 \\ 0 & Z_{i,j} & Y_{i,j}^- & 0 \\ F_{i,j} & 0 & 0 & X_{i,j}^- \end{pmatrix}. \tag{23}$$

Finally, the concurrence is given by

$$C = \max\{0, C_1, C_2\}, \qquad C_1 = 2\left(|Z_{i,j}| - \sqrt{X_{i,j}^+ X_{i,j}^-}\right), \qquad C_2 = 2\left(|F_{i,j}| - \sqrt{Y_{i,j}^+ Y_{i,j}^-}\right). \tag{24}$$

It is important to note that the elements of the reduced density matrix for nearest-neighbor spin pairs are obtained as

$$X_{n,n+1}^+ = f_0^2 - |f_1|^2 + |f_2|^2, \quad X_{n,n+1}^- = 1 - 2f_0 + X_{n,n+1}^+, \quad Y_{n,n+1}^+ = Y_{n,n+1}^- = f_0 - X_{n,n+1}^+, \\ Z_{n,n+1} = f_1, \qquad\qquad F_{n,n+1} = f_2, \tag{25}$$

where

$$\begin{aligned} f_0 &= \frac{1}{N}\sum_{k\in\Lambda}\cos^2(\theta_k), & \varepsilon(k) &< 0, \\ &= \frac{1}{N}\sum_{k\notin\Lambda}\sin^2(\theta_k), & \varepsilon(k) &> 0, \\ f_1 &= \frac{1}{N}\sum_{k\in\Lambda}\left[\cos(k)\cos^2(\theta_k) - i\sin(k)\right], & \varepsilon(k) &< 0, \\ &= \frac{1}{N}\sum_{k\notin\Lambda}\cos(k)\sin^2(\theta_k), & \varepsilon(k) &> 0, \\ f_2 &= -\frac{i}{2N}\sum_{k\in\Lambda}\sin(2\theta_k)\sin(k), & \varepsilon(k) &< 0, \\ &= \frac{i}{2N}\sum_{k\notin\Lambda}\sin(2\theta_k)\sin(k), & \varepsilon(k) &> 0. \end{aligned} \tag{26}$$

QD is a measure of non-classical correlations between two subsystems of a quantum system [51]. Unlike entanglement, which only captures a specific type of quantum correlation, QD includes all types of quantum correlations, even those present in separable (non-entangled) states. QD is defined as the difference between two expressions of mutual information in a quantum system. In classical information theory, mutual information is a measure of the total correlations between two systems. However, in the quantum realm, there are two different ways to generalize this concept, leading to the definition of QD.

Mathematically, for a two particles the QD is given by:

$$QD = I(\rho_{ij}) - J(i|j), \tag{27}$$

where, $I(\rho_{ij})$ is the total mutual information, defined as:

$$I(\rho_{ij}) = S(\rho_i) + S(\rho_j) - S(\rho_{ij}). \tag{28}$$

Here, $S(\rho)$ denotes the von Neumann entropy of the state $\rho$. $J(i|j)$ represents the classical correlations, defined as:

$$J(i|j) = S(\rho_i) - \min_{\Pi_m^j} \sum_m p_m S\left(\rho_{i|\Pi_m^j}\right),\tag{29}$$

where $\Pi_m^j$ is a set of projective measurements on subsystem $j$, and $p_m$ is the probability of outcome $m$.

We have calculated the concurrence and QD between nearest neighbor spin pairs, with the results presented in Fig. 3 (a) and (c). A chain size of $N = 1000$ and $\alpha = 1$ were considered. At the quantum critical point, $\beta_c = -1$, both concurrence and QD reach their maximum values. Additionally, as shown in the insets of Fig. 3 (a) and (c), the first derivatives of these quantities with respect to $\beta$ exhibit a jump at the quantum critical point. This jump indicates a significant change in the system's quantum correlation properties and can signal a continues quantum phase transition. In such transitions, the first derivative of the order parameter (such as concurrence) with respect to the control parameter is continuous, while the second derivative is discontinuous. The observed fluctuations in the region where $\beta < \beta_c$ are due to the size effect on the first derivatives.

Interestingly, Fig. 3 (a) reveals two entangled points in two gapless phases, which we denote as $\beta_E^\pm$. In the gapless chiral phase, nearest-neighbor spin pairs are not entangled as $\beta \longrightarrow -\infty$. As $\beta$ increases, these spin pairs remain unentangled until the entangled anisotropy $\beta_E^-$ is reached. Beyond this point, as anisotropy increases further, the nearest-neighbor spin pairs become entangled, and concurrence rapidly grows until the quantum critical point, $\beta_c$. Conversely, in the gapless chiral-nematic phase, no entanglement is observed between nearest-neighbor spin pairs for $\beta > \beta_E^+$. However, as soon as the anisotropy decreases from $\beta_E^+$, entanglement is created between these spin pairs. This entanglement increases gradually with decreasing $\beta$, reaching its maximum at the quantum critical point.

The vanishing concurrence observed at specific points within the gapless phases, such as $\beta_E^\pm$, presents an intriguing phenomenon, as it does not align with any known phase transition lines. This behavior can be understood as a result of the competition between quantum fluctuations and the ATSI, which shapes the entanglement properties of the system. Physically, when $\beta$ approaches the unentangled regions near $\beta_E^\pm$, the interaction strength between nearest-neighbor spins becomes insufficient to sustain quantum correlations, largely due to the influence of local quantum fluctuations. These fluctuations disrupt the entanglement by minimizing the overlap between spin states required for concurrence, leading to its suppression in these regions. Remarkably, this absence of concurrence underscores a regime where nearest-neighbor spins are less correlated, governed by the subtle balance between ATSI interactions and quantum fluctuations. Although these points of vanishing concurrence are not directly tied to phase transition lines, they act as thresholds beyond which entanglement either emerges or disappears, driven by the system's anisotropy and quantum critical properties. Consequently, these points reflect the nuanced nature of quantum correlations and offer valuable insight into the dependence of entanglement on system parameters.

The entangled region is significant for several reasons. In quantum information science, entanglement is a crucial resource for quantum computing, allowing quantum bits (qubits) to perform complex computations more efficiently than classical bits. Entangled states are also utilized in quantum communication protocols, such as quantum key distribution, which provides secure communication channels that are theoretically immune to eavesdropping. Additionally, entanglement enhances the precision of measurements in quantum sensing and imaging technologies, leading to applications in high-resolution medical imaging. Understanding entanglement in condensed matter systems can also pave the way for developing new materials with unique properties, such as high-temperature superconductors.

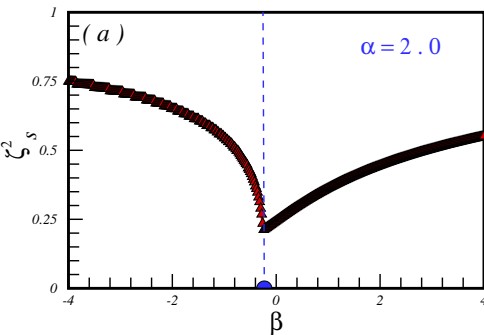
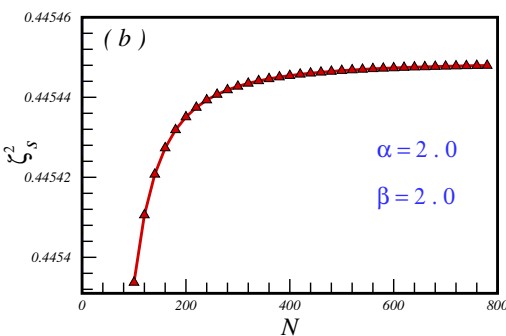

Figure 4: (color online) (a) The behavior of the SSP as a function of $\beta$ at $\alpha = 2.0$ for a chain system with $N = 1000$ particles. (b) The SSP as a function of the chain size $N$ at $\alpha = 2.0$ and $\beta = 2.0 > \beta_c$.

Fig. 3 (b) and (d) present the density plots of concurrence and QD as functions of $\alpha$ and $\beta$ for a chain size of $N = 1000$. In both plots, the critical line defined by $\beta = -1/\alpha^2$ is clearly observed, indicating the boundary between different quantum phases. Additionally, it is evident that as $\alpha$ increases, the width of the entangled region decreases. This suggests that the range of $\beta$ values over which nearest-neighbor spin pairs remain entangled becomes narrower with increasing $\alpha$. This behavior highlights the sensitivity of the entangled region to changes in the system's parameters, providing deeper insights into the interplay between $\alpha$ and $\beta$ in determining the entanglement properties of the system.

### 3.3 The spin squeezing and entanglement entropy

Spin squeezing is a quantum phenomenon that reduces the variance of one component of angular momentum in a collection of spin particles [52–55]. This reduction creates spin-squeezed states, which are highly valuable in quantum metrology and quantum information science. These states can improve measurement precision beyond the standard quantum limit, making them particularly useful for tasks like estimating rotation angles more accurately than classical interferometers. Additionally, spin squeezing plays a crucial role in quantum phase transitions [56–68].

To quantify spin squeezing, we use the spin squeezing parameter (SSP), which provides a measure of the degree of squeezing in the system. The SSP, denoted as $\xi_s^2$ and is defined as [52, 68]

$$\xi_s^2 = \frac{2}{N} \left[ \left\langle J_x^2 + J_y^2 \right\rangle - \sqrt{\left\langle J_x^2 - J_y^2 \right\rangle^2 + \left\langle J_x J_y + J_y J_x \right\rangle^2} \right], \tag{30}$$

where $J_\nu = \sum_{n=1}^{N} S_n^\nu$ for $\nu = x, y, z$ are total spin operators. A state is considered spin-squeezed if $\xi_s^2 < 1$. This condition indicates that the variance in one component of the angular momentum is reduced below the standard quantum limit, signifying the presence of quantum entanglement among the particles. Additionally, $\xi_s^2 = 1$ corresponds to a coherent state, which represents a minimal uncertainty state, exhibiting maximum coherence and classical behavior.

In Fig. 4, we present our findings for the SSP. In Fig. 4 (a), the SSP is plotted against the anisotropy parameter $\beta$ for a chain size of $N = 1000$ and $\alpha = 2$. It is evident that the ground state of the system exhibits squeezing across the entire range of the ground state phase diagram. Notably, at the critical point $\beta_c = -0.25$, the ground state becomes extremely squeezed, and a distinct signal is observed. This indicates a significant change in the quantum correlations at the critical point.

Furthermore, in the gapless chiral-nematic phase, where $\beta > \beta_c$, the ground state shows even more pronounced squeezing compared to the chiral phase region where $\beta < \beta_c$. This increased squeezing in the gapless phase suggests stronger quantum correlations and entanglement in this region. The behavior of the SSP across different phases provides valuable insights into the nature of quantum correlations and the entanglement structure in the system, highlighting the critical role of anisotropy in determining the ground state properties. These findings contribute to a deeper understanding of the quantum phase transitions and the entanglement characteristics in spin-1/2 chains.

In the following section, we explore how the SSP changes with variations in system size. Investigating the scaling behavior of SSP is essential for several reasons. By analyzing the scaling behavior of SSP, we can gain a deeper understanding of quantum phase transitions and the associated critical phenomena. Additionally, examining SSP's scaling behavior can enhance quantum sensors and measurement devices, achieving higher precision close to the Heisenberg limit.

In the field of quantum metrology and quantum sensing, the precision of quantum measurements in spin systems is determined by two fundamental limits: the standard limit and the Heisenberg limit. The standard limit, also known as the shot-noise limit or standard quantum limit, defines the precision achievable with uncorrelated or coherent spin states [69, 70]. This limit scales as $\zeta_s \propto 1/\sqrt{N}$, where $N$ represents the number of spins. Essentially, as the number of spins increases, the precision improves, but only up to a certain point dictated by this limit.

In contrast, the Heisenberg limit, also referred to as the ultimate limit or quantum Cramér-Rao bound, represents the precision attainable with entangled or squeezed spin states. This limit scales as $\zeta_s \propto 1/N$ [71–73], indicating a much higher precision compared to the standard limit. The Heisenberg limit sets the highest possible precision achievable by any quantum state, surpassing the standard limit by a factor of $1/\sqrt{N}$. This means that for a large number of spins, the precision at the Heisenberg limit is significantly better than at the standard limit.

We present our scaling results in Fig. 4 (b). It is important to note that we observed scaling behavior only in the gapless region where $\beta > \beta_c$. Therefore, the results in this figure are provided for $\alpha = 2.0$ and $\beta = 2.0 > \beta_c = -0.25$. As illustrated in Fig. 4 (b), the SSP scales as $\zeta_s^2 = a + b/N^2$.

Surprisingly, we found that in the gapless chiral-nematic phase, the Heisenberg limit can be achieved. This is a significant finding as it suggests that the system exhibits a high degree of quantum correlations, allowing for precision measurements that approach the ultimate quantum limit. Achieving the Heisenberg limit in this phase implies that the system can be used for highly sensitive quantum sensing and metrology applications, where the precision of measurements is crucial. This result highlights the potential of the chiral-nematic phase for practical applications in quantum technologies, making it an exciting area for further research and exploration.

Spin squeezing and entanglement entropy (EE) [74] are closely related concepts in quantum mechanics and both of them are invaluable tools for studying the ground state phase diagram of quantum systems. EE measures the degree of quantum entanglement between two subsystems of a composite quantum system. It quantifies the information loss or inaccessibility when only one of the subsystems is accessible, rather than the entire system.

EE is crucial in studying the ground state phase diagram because it provides deep insights into the quantum entanglement properties of a system. By measuring the degree of entanglement between different parts of a system, EE helps identify and characterize quantum phase transitions and critical points. Changes in EE can signal different quantum phases and their boundaries, offering a clear picture of the underlying quantum phenomena. This makes EE a powerful tool for understanding the complex behavior of quantum systems and mapping out

their ground state phase diagrams.

One common method to define EE involves the density operator and calculating the von Neumann entropy of a reduced density matrix for a subsystem [75,76]. For a bipartite system, the EE is defined as the von Neumann entropy of subsystem $A$:

$$S_A = -\mathbf{Tr}\left[\rho_A \log_2(\rho_A)\right],\tag{31}$$

where $\rho_A$ is the reduced density matrix of $A$, obtained by tracing over the rest of the system, $B$:

$$\rho_A = \mathbf{Tr_B}(\rho).\tag{32}$$

EE typically scales with the boundary area of subsystem $A$, rather than its volume. This behavior, known as the 'area law' for EE, has been extensively studied. In noncritical ground states of spin chains with a finite correlation length, the EE remains constant. However, at a quantum critical point, when subsystem $A$ is a finite interval of length $l$, the EE deviates slightly from the area law due to a logarithmic correction:

$$S_A(l) \sim \frac{c_{eff}}{3}\log(N),\tag{33}$$

where $c_{eff}$ represents the central charge [77–79].

To gain deeper insights into the various gapless phases of the system, we calculated the EE in chain systems with different Hamiltonian parameters. We divided the system into two equal parts, where $N_A = N_B = \frac{N}{2}$. The results are presented in Fig. 5 for $N = 1000$ and $\alpha = 2$. As shown in Fig. 5 (a), in the chiral gapless phase where $\beta < \beta_c = -0.25$, the EE is very large and increases rapidly as we move away from the critical point. In contrast, in the other

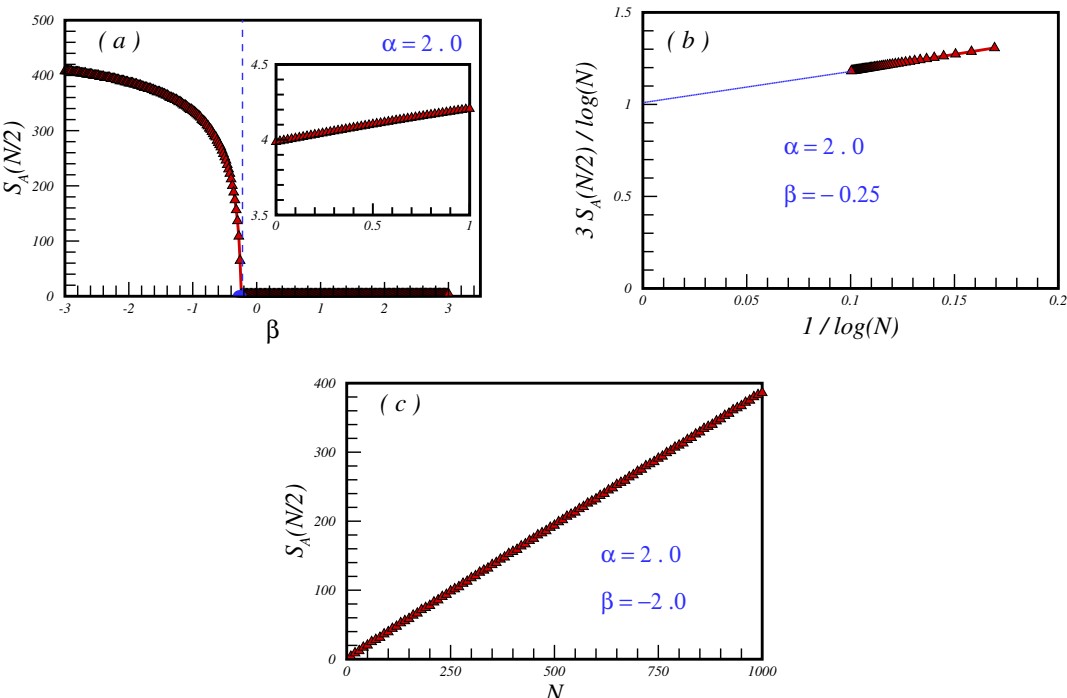

Figure 5: (color online) (a) The EE as a function of $\beta$ at $\alpha = 2.0$ for a chain with $N = 1000$ particles. (b) A linear fit of the form $Y = mX + b$ applied to the function $3S_A(N/2)/\log(N) = c_{eff} + m/\log(N)$ on the critical line $\beta_c = -0.25$ at $\alpha = 2.0$. (c) The EE in the gapless chiral phase as a function of the chain size $N$.

gapless region where $\beta > \beta_c = -0.25$, the two equal parts are weakly entangled compared to the chiral gapless phase, and the EE remains almost constant with respect to $\beta$. It is important to note that the EE reaches its minimum value at the critical point $\beta = \beta_c = -0.25$.

This behavior highlights the distinct entanglement properties in different gapless phases and underscores the critical point as a unique feature where the EE is minimized. These findings provide valuable information about the quantum correlations present in the system and can help in understanding the nature of the gapless phases more comprehensively.

In the final step, we examined the scaling behavior of EE with respect to the system size. This analysis is crucial for several reasons. Firstly, it serves as a powerful tool to differentiate between various quantum phases. At critical points, EE often displays unique scaling properties that are distinct from those in bulk phases. By investigating how EE scales with system size, researchers can gain valuable insights into the distribution and strength of quantum correlations in the ground state, which is particularly important for understanding the entanglement structure in strongly correlated systems. Additionally, the scaling of EE has significant implications for the efficiency of classical simulations of quantum systems. Systems exhibiting area law scaling of EE can often be efficiently simulated using tensor network methods, such as matrix product states, whereas systems with volume law scaling of EE are generally more challenging to simulate classically. Overall, studying the scaling of EE in spin-1/2 chain systems provides a deep understanding of the quantum properties and phase structure of these systems, making it a vital aspect of modern condensed matter physics and quantum information science. Results are presented in Fig. 5 (b) and (c).

Fig. 5 (b) presents the results when the system is at the quantum critical point $\beta = \beta_c = -0.25$. Our findings reveal that the central charge in this scenario is $c_{\text{eff}} = 1.0$. We conducted similar calculations for all critical points along the critical line $\beta_c = -\frac{1}{\alpha^2}$ and consistently found the same central charge. Different values of the central charge indicate distinct critical behaviors and symmetries of the system at these points. For instance, a central charge of 0.5 is often linked to systems like the transverse field Ising model at criticality, 1.0 is typical for the isotropic XX spin-1/2 chain. Additionally, we observed that when the system is in the gapless chiral-nematic phase, $\beta > \beta_c$, the logarithmic behavior with a central charge of $c_{\text{eff}} = 1.0$ indicates that this phase is also critical.

Interestingly, we have found that the EE in the gapless chiral phase exhibits a completely different scaling behavior. As shown in Fig. 5 (c), EE in the region $\beta < \beta_c$ displays a linear relationship with the system size, indicating that EE is an extensive parameter and follows volume-law scaling, which is not typical for gapped systems but can occur in critical systems or those with long-range entanglement [80, 81].

This behavior indicates that the system has a high degree of quantum correlations spread across its entire length, rather than being localized near the boundaries. This finding is significant as it provides insights into the nature of the quantum state in the gapless chiral phase, highlighting the presence of strong, long-range entanglement. Understanding this behavior is crucial for developing efficient simulation methods and for gaining a deeper comprehension of the entanglement structure in strongly correlated quantum systems.

Volume-law scaling of EE in a spin-1/2 chain has several important applications across various fields. In condensed matter physics, it helps in understanding the nature of quantum phases and phase transitions, particularly in systems with high entanglement. This scaling provides insights into the behavior of strongly correlated materials and can reveal complex topological features and multiple degrees of freedom. In quantum information science, volume-law scaling is crucial for studying quantum chaos and integrability. It aids in diagnosing the entanglement structure of quantum states, which is essential for developing efficient quantum algorithms and understanding the limits of quantum computation.

In quantum metrology, volume-law scaling can enhance the precision of measurements by exploiting the extensive entanglement in quantum systems. This is particularly useful in achieving better-than-classical precision in estimating parameters, such as magnetic fields or gravitational waves, through quantum-enhanced sensing techniques. Additionally, in quantum measurements, the study of volume-law scaling can lead to a deeper understanding of measurement-induced phase transitions. It helps in identifying the conditions under which a system transitions from volume-law to area-law entanglement, which is important for optimizing measurement protocols and improving the accuracy of quantum state tomography. These applications highlight the significance of volume-law scaling in advancing our understanding and capabilities in various domains of quantum science and technology.

# 4 Conclusion

Our comprehensive analysis of the spin-1/2 XX chain model with anisotropic three-spin interaction has revealed a rich tapestry of ground state properties and phase behaviors. By employing the fermionization technique and exact diagonalization of the Hamiltonian, we have identified two distinct gapless phases separated by a Lifshitz critical line. The gapless phase is characterized by chiral long-range ordering, while the chiral-nematic phase exhibits unique long-range ordering.

In the first part of our study, we analyzed the dispersion relation, ground state energy, and order parameters. These analyses allowed us to map out the phase diagram and identify the Lifshitz critical line separating the two gapless phases. The phase, with its chiral ordering, and the chiral-nematic phase, with its distinct ordering, highlight the complex interplay of interactions in the system.

Next, we focused on the quantum correlations between nearest-neighbor spins by examining the concurrence and QD. Our findings indicate that these measures are maximized at the critical line, underscoring the enhanced quantum correlations in this region. Additionally, we observed an entangled region, further emphasizing the intricate quantum nature of the ground state.

In the final part of our study, we investigated the SSP and EE. Our results show that the ground state is squeezed throughout the phase diagram, with extreme squeezing occurring at the Lifshitz critical line. Remarkably, in the gapless chiral-nematic phase, the Heisenberg limit is achieved, indicating the potential for high-precision measurements in this phase. By dividing the system into two equal parts, we found significant entanglement in the gapless chiral phase, highlighting the strong quantum correlations present.

The central charge calculation confirms the critical nature of the gapless chiral-nematic phase, based on the scaling of the EE. Furthermore, the EE follows volume-law scaling in the gapless chiral phase, indicating that it is an extensive parameter in this region.

The unconventional entanglement observed in the gapless chiral phase, as indicated by volume-law scaling of the EE, can be understood through the unique characteristics of the system. Unlike the standard logarithmic scaling derived from conformal field theory, which reflects localized quantum correlations, the extensive entanglement suggests the presence of long-range quantum correlations spreading across the entire system. This behavior could be attributed to the ATSI in the model, which induces strong quantum correlations beyond typical conformal field theory descriptions. These phenomena highlight the ground state's complex nature and suggest that its entanglement structure transcends conventional frameworks, requiring further investigation into the mechanisms governing this behavior.

The deviation from the standard Calabrese-Cardy formula may indicate that the gapless composite phase is not fully described by traditional conformal field theory. While conformal field theory relies on conformal symmetry and standard universality, the observed volume-law scaling reflects the intricate interplay of anisotropic interactions, quantum correlations, and critical properties, suggesting the need for alternative theoretical frameworks. The Jordan-Wigner transformation could introduce emergent features, such as intertwined degrees of freedom or unconventional scaling, which are not accounted for by conformal symmetry. These findings challenge the notion of universal logarithmic scaling in gapless systems and emphasize the importance of extending current theoretical models to capture the rich entanglement structure of strongly correlated phases. This opens up an exciting avenue for exploring nonstandard critical phenomena and entanglement scaling in quantum systems.

Overall, our findings contribute to the understanding of quantum phase transitions and the role of multi-spin interactions in low-dimensional quantum systems. The identification of distinct gapless phases, critical lines, and regions of enhanced quantum correlations provides valuable insights for future research in quantum information and condensed matter physics. These results may also have implications for the development of quantum technologies, where understanding and harnessing quantum correlations and entanglement are crucial.

## Acknowledgments

We sincerely thank M. Mohammadi Mozafar for reading the section on entanglement entropy and for the insightful discussions and valuable comments on this topic.

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
