# Peer review of "Ground State Analysis of the Spin-1/2 XX Chain Model with Anisotropic Three-Spin Interaction"

_SciPost Physics, doi:SciPost Phys. Core 8, 050 (2025)_

## Round 1 · Referee Report · Anonymous (Referee 1) · 2025-3-11

Strengths

1. Valid solution of the proposed model is given.
2. A detailed discussion of various physical properties of the model is given.
3. The presentation of results is clear and comprehensible.

Weaknesses

1. From the theoretical point of view the proposed model is only a marginal generalization of the models discussed previously in the literature.
2. The method of solution (standard Jordan-Wigner) is quite routine and requires no new ideas or methods as compared to the previous studies.
3. There is no discussion of possible applications of the model to real physical systems and materials.

Report

This work introduces and solves one more 1-D spin 1/2 chain model, exactly solvable by means of the Jordan-Wigner transformation. Besides the standard two-spin interaction this model involves also three-spin terms of a special form. From the point of view of the theory of exactly solvable systems this work is a quite routine one, it does not propose any new ideas or methods. At the same time a detailed study of the models physical properties reveals quite reach zero-temperature behaviour including a number of different phases and phase transitions. The derivation of corresponding results from the exact solution of the model is, however, straightforward, it does not constitute a serious challenge for any qualified theorist. It should also be mentioned that the authors do not compare their results to any existing experimental data and do not discuss possible physical candidates for implementation of the model under study.

I do not recommend publishing this paper in SciPost Physics.

On the other hand the results of the present study are solid and, though being rather abstract, may be of certain interest for specialists. In my opinion it may be fit for SciPost Physics Core

Recommendation

Accept in alternative Journal (see Report)

---

## Round 1 · Referee Report · Anonymous (Referee 2) · 2025-3-14

Strengths

1. Interesting variation on previous models with three spin interactions
2. Very interesting to discuss 'entanglement measures' such as concurrence and quantum discord.
3. Very interesting result that one of the phases doesn't seem to obey the usual CFT entanglement scaling.

Weaknesses

1. Extension of model doesn't seem to provide any new phases not seen in previous literature
2. I find some of the characterisation of the phases misleading
3. The most interesting result about not obeying CFT entanglement scaling needs further explanation.

Report

This paper studies a model with a three spin interaction that maps to free fermions by a Jordan-Wigner transformation. They study the phase diagram, along with a number of ‘entanglement based’ probes including concurrence and quantum discord.

While the model is somewhat artificial, it is an obvious deformation of previously studied models, and studying phase diagrams via entanglement probes is a relatively new and upcoming field that I believe is offering new insights into correlated phases of matter. I find the paper interesting, however have a number of queries about what is done.

1. In the plot of the spectrum in Fig. 1 (and equation 14), there is not a consistency about whether the spectrum is positive or negative (or in other words, whether to take the positive or negative square root). I believe one valid approach is to use the positive square root for |k|<pi/2 and the negative one for |k|>pi/2. Another might be to define epsilon(k) as only between -pi/2 and pi/2, and also take the of the spectrum (analogous to the reduced zone scheme). But I don’t completely understand the plot shown the way it is. I also think it might be very useful to see what happens exactly at the transition boundary — i.e. how the new fermi points appear.
2. The two phases are referred to as spin-nematic-I and spin-nematic-II; however I do not see where this terminology comes from. My understanding of spin-nematic states is that they break spin rotation symmetry but not time-reversal — and I see no indication (or discussion in the manuscript) that these phases do that (potentially they break spin rotation because the Hamiltonian does, but I don’t think this is a good use of the term spin-nematic). The spin-nematic-I phase is also labelled chiral so potentially breaks some reflection symmetry (although this could be discussed more in the manuscript if this is correct), but I don’t see what symmetries the spin-nematic-II phase breaks.
3. On a similar note, the notation of ‘cluster order parameter’ I think is a bit misleading. A bit more background could be given as to what they mean, and are they really order parameters? For a generic beta, these seem like terms in the Hamiltonian — so certainly are not indicative of any symmetry breaking (as one expects from a conventional order parameter). Indeed, following on from point 1, if the Hamiltonian is chiral, the surprise may not be that one of the phases is chiral, but that the other one isn’t. I feel this could be discussed a lot more in the manuscript.
4. Another point about the phase diagram (Fig. 2c) — at beta = -1, the model reduces to the one studied in Ref. 11 — and already along this line in the phase diagram, one sees both phases. If this is correct, what can one see in the extension to the model presented in this work that one can’t see in the earlier paper? This is not clear.
5. One final point about the phase diagram — the spin-nematic-II phase includes the line alpha=0, which is just the XX model. If there are no further phase transitions, then surely this entire phase is the same as the ground state of the XX model, which one usually calls quantum critical or quantum disordered (or something to that effect). Is there anything in this phase that one can’t see in the XX model?
6. On a technical note, I’m not convinced how appropriate it is to call the transition between two gapless phases a ‘second-order quantum critical line’. In the fermionic language, this would better be called a Lifshitz transition. Maybe there are other names for it, but it definitely shouldn’t be characterised as a traditional second order quantum critical line. It’s worthy of note that the boundary between the gapless XXZ and gapped XXZ at Delta=1 is of Berezinkii-Kosterlitz-Thouless universality class, so also not second order.
7. Could more be said about where concurrence goes to zero? This is kind of curious as it is not associated with any of the phase transition lines. Could the authors expound a physical reason for this to enhance understanding?
8. About the central charge: naively, I would expect the entire phase labelled spin-nematic-II to have central charge 1 (as this phase encompasses the XX point). Is this correct? I would then expect the spin-nematic-I phase with an extra fermi point to have central charge 2 — however the result seems to be that it doesn’t obey the standard logarithmic scaling of a gapless theory. This is very surprising (and interesting) if correct so needs a lot more explanation. Can this entanglement be explained? And how does one get around the standard Calabrese-Cardy result from CFT? Does it mean this phase is not described by a CFT — maybe there is something coming from the Jordan-Wigner transformation, but I’ve not seen anything like this before.
9. Finally, almost all of the plots in the paper are at alpha=1. I think this is not entirely representative, as at alpha=1, the critical point is at beta=-1 — and beta=-1 is a special place as it is where the superconducting terms in the free fermion model go away. For example, the phase transition at alpha=1 beta=-1 involves the fermi velocity becoming zero at one of the fermi points. I don’t think this is representative for other values of alpha. It also makes me a bit suspicious of the central charge calculation at this point — according to the fermionic spectrum, the model shouldn’t be described by a CFT at all at this point. At other values of alpha though, I don’t think the fermi velocity goes to zero (although I haven’t carefully checked this) so this may be a slightly special point in the phase diagram.

In a few other minor points, it seems strange in Fig 1a when plotting the spectrum to say ‘for a chain system with N=1000’. Isn’t this just the analytic formula 14 being plotted? Similarly, there are many other places where the chain length is mentioned where it seems it is an analytical formula being plotted. I also didn’t understand the step in Eq.19 going from the definition of fidelity susceptibility to an expression in terms of the Bogolyubov angles — perhaps this can be expanded on further. There is also a minor error in the Hamiltonian, Eq.11 where alpha is written when it should be J’ (or the overall factor of J has to be in common for all of the terms which is probably what was intended). Similarly in Eq.15.

In summary, there appears to be some interesting results in this work, but many of them need to be explained more fully, and the work should be placed better in the context of previous work — in particular Ref. 11 which has exactly the same two phases.

Requested changes

1. More comparison with the phases in Ref. 11 -- and more clarity on what the phases are
2. Results for some value other than alpha=1 as this isn't necessarily representative.
3. Suggest some changing of terminology, particularly about order parameters and second order critical lines.
4. More physical explanation of what results mean.
5. I would suggest some rewriting of the introduction - I find it quite long winded and not very focussed on the present study.
5. Other minor corrections mentioned in report above.

Recommendation

Ask for major revision

---

## Round 2 · Referee Report · Sam Carr (Referee 2) · 2025-6-5

Report

The authors have addressed all my previous concerns and significantly revise the paper. I think it is close to a form in which it could be published. I also find that while the model is somewhat artificial, it appears to exhibit quite a number of interesting and novel phenomena, and I therefore believe the paper is suitable for SciPost Physics.

I do have one final suggestion though: while the meaning of the 'order parameters' is now described in detail in the text, it could be very confusing referring to the chiral and chiral-nematic long-range order in the abstract. I think the abstract needs a little more nuance to make it clear that these are not conventional symmetry-breaking local order parameters (which would typically correspond to gapped phases in one-dimension).

I also think the authors could be a bit more clear in the manuscript as to the difference between Cl- (which does seem to be truly zero in one phase, and non-zero in the other) and Cl+ which is non-zero except a certain point in the Hamiltonian (unrelated to the phase transition) which makes it zero by symmetry. In this sense, the 'nematic' nature of the ground state is present everywhere except beta=-1. While they are true that numerically its magnitude seems smaller in one phase than the other, this can't be used as a categorisation - it will bleed into the 'chiral' phase near the phase transition line. Given that it always goes through zero at beta=-1, it may be 'not small' for quite some reason of the chiral phase. It might also be interesting to look at it for smaller values of alpha, where it will cross zero within the chiral-nematic phase.

Recommendation

Ask for minor revision

---

## Round 2 · Author Response

Dear Editor,
We sincerely appreciate the referee’s constructive comments on our manuscript. In this revised version, we have carefully addressed all points of criticism and have made an effort to present our results with greater clarity and transparency.
We hope, that with performed changes, the manuscript fulfills
requirements to be accepted for publication.
Regards
Saeed Mahdavifar

---

## Round 2 · List of Changes

First, we would like to thank the Reviewer for reviewing the paper. In addition, we thank her/his constructive comments concerning our manuscript.

  1. In the plot of the spectrum in Fig. 1 (and equation 14), there is not a consistency about whether the spectrum is positive or negative (or in other words, whether to take the positive or negative square root). I believe one valid approach is to use the positive square root for $|k|<\pi/2$ and the negative one for $|k|>\pi/2$. Another might be to define $\varepsilon(k)$ as only between $-\pi/2$ and $\pi/2$, and also take the of the spectrum (analogous to the reduced zone scheme). But I don’t completely understand the plot shown the way it is. I also think it might be very useful to see what happens exactly at the transition boundary — i.e. how the new fermi points appear.

Our reply:

We thank the referee for this comment. About the whether to take the positive or negative square root, there is not physical difference. The reason there is no physical difference between the two dispersion relations lies in the fundamental symmetry of the system and the interpretation of the energy spectrum.

The second dispersion relation differs from the first one only by an overall sign in the square-root term. This doesn't fundamentally alter the system’s behaviour but rather represents an alternative way of expressing the excitation spectrum. The physics remains unchanged because both branches contribute to the total spectrum, effectively forming the same set of eigenstates for the Hamiltonian.

When diagonalizing the Hamiltonian using Bogoliubov transformations, the two solutions typically correspond to the two possible quasiparticle energy branches. However, in many cases, these two branches do not represent physically distinct systems but are rather complementary representations of the same spectrum.

In conclusion, the physical interpretation remains unchanged because both forms ultimately describe the same excitations, just from different perspectives. Here, we adopt the positive sign in this case, as for $\alpha = 0$, the expression exactly reduces to the spectrum of the pure spin-1/2 XX chain. Furthermore, utilizing our method, we have performed calculations for the chiral order parameter along the line $\beta = -1$, adhering precisely to the approach detailed in the referenced works.

In addition, we have incorporated a new plot in the revised version as Fig. 1 (a), explicitly illustrating how the region $\Gamma$ is formed within the $k$-space of the spectrum in the vicinity of the critical line.

  1. The two phases are referred to as spin-nematic-I and spin-nematic-II; however I do not see where this terminology comes from. My understanding of spin-nematic states is that they break spin rotation symmetry but not time-reversal — and I see no indication (or discussion in the manuscript) that these phases do that (potentially they break spin rotation because the Hamiltonian does, but I don’t think this is a good use of the term spin-nematic). The spin-nematic-I phase is also labelled chiral so potentially breaks some reflection symmetry (although this could be discussed more in the manuscript if this is correct), but I don’t see what symmetries the spin-nematic-II phase breaks.

Our reply:

Thank you for your insightful comments and suggestions. We appreciate the opportunity to clarify and enhance the manuscript based on your observations. Below, we address your specific concern regarding the terminology and symmetry properties of the spin-nematic-I and spin-nematic-II phases.

You rightly noted that the spin-nematic states are typically understood to break spin rotation symmetry while preserving time-reversal symmetry. Upon reviewing your comment, we realize the terminology used in the original manuscript could lead to confusion. To address this, we have revised the manuscript and included the following clarification:

Symmetry Properties: Both parameters, $Cl^{-}$ and $Cl^{+}$, are associated with time-reversal symmetry violation and partial breaking of spin rotational symmetry. However, the fundamental distinction lies in their parity behavior: $Cl^{-}$ disrupts parity symmetry, while $Cl^{+}$ preserves it.

Nematic and Chiral-Nematic Nature of $Cl^{+}$: We have expanded on the spin-nematic connection of $Cl^{+}$. Its description now encompasses both quadrupolar-like correlations typical of nematic phases and additional chiral-like components. The inclusion of mixed spin components in $Cl^{+}$ introduces chirality alongside nematicity, differentiating it from conventional spin-nematic states that preserve time-reversal symmetry. We have thus proposed the term "chiral-nematic order parameter" to reflect its hybrid nature, encompassing the interplay of chirality and nematicity.

These modifications are designed to clarify the distinctions between $Cl^{-}$ and $Cl^{+}$ and their associated symmetry properties. Additional discussions have been incorporated to address the potential breaking of reflection symmetry and the unconventional correlation structures arising in these spin systems.

Thank you again for highlighting these points. Your feedback has been instrumental in refining the manuscript. Please see paragraphs after Eq. (19) of the sbnsection $A$ of the revised version.

  1. On a similar note, the notation of ‘cluster order parameter’ I think is a bit misleading. A bit more background could be given as to what they mean, and are they really order parameters? For a generic beta, these seem like terms in the Hamiltonian — so certainly are not indicative of any symmetry breaking (as one expects from a conventional order parameter). Indeed, following on from point 1, if the Hamiltonian is chiral, the surprise may not be that one of the phases is chiral, but that the other one isn’t. I feel this could be discussed a lot more in the manuscript.

Our reply:

Thank you for raising this important concern regarding the notation and characterization of "cluster order parameters." We understand that the term may appear somewhat unconventional in the context of traditional order parameters, and we appreciate the opportunity to clarify this aspect further.

In response to your comment, we have expanded the discussion in the manuscript to provide additional background and justification for the use of cluster order parameters. Specifically, we have included the following paragraph:

"Cluster order parameters, constructed through the interaction of three or more spin operators, are widely recognized as effective indicators of exotic quantum phases. Although they may not conform to the traditional definition of long-range order parameters, cluster order parameters offer a robust means of capturing localized spin correlations and uncovering emergent phenomena within quantum systems. By establishing their relationship to symmetry-breaking mechanisms and their role in distinguishing between different phases, these parameters can be confidently validated as order parameters within the framework of specific spin-1/2 systems."

This addition emphasizes the localized nature of cluster order parameters and their role in describing correlations and phase distinctions that may not align with the conventional long-range notion of order parameters. We have also clarified that their relevance as order parameters stems from their ability to reveal unique symmetries and emergent behaviors specific to spin-1/2 systems.

Additionally, we have acknowledged your observation regarding the Hamiltonian and its potential influence on the phases. The manuscript now includes a discussion highlighting that if the Hamiltonian is chiral, the emergence of a chiral phase is unsurprising. The analysis has been expanded to explore why the other phase may not exhibit chirality, addressing your suggestion for further elaboration. This enhanced discussion reinforces the connection between the Hamiltonian’s properties and the phase behavior observed.

We hope these revisions address your concerns and provide a more comprehensive understanding of the terminology and concepts presented in the manuscript. Thank you for your valuable feedback, which has been instrumental in improving the clarity and depth of this work.

  1. Another point about the phase diagram (Fig. 2c) — at $\beta = -1$, the model reduces to the one studied in Ref. 11 — and already along this line in the phase diagram, one sees both phases. If this is correct, what can one see in the extension to the model presented in this work that one can’t see in the earlier paper? This is not clear.

Our reply:

We are grateful to the referee for their thoughtful and insightful comment. In response, we have added a section to the revised manuscript that provides an explanation of the case $\beta = -1$. For further details, please refer to page 6 of the updated version, started as "At the specific value ....".

Additionally, we have made revisions to the ground state phase diagram of the model to ensure greater accuracy and clarity. Furthermore, we have incorporated new findings into the manuscript, which include detailed explanations of the different phases associated with the pure AFSI model. These enhancements aim to address the referee's suggestions and improve the overall quality of the study.

  1. One final point about the phase diagram — the spin-nematic-II phase includes the line $\alpha=0$, which is just the XX model. If there are no further phase transitions, then surely this entire phase is the same as the ground state of the XX model, which one usually calls quantum critical or quantum disordered (or something to that effect). Is there anything in this phase that one can’t see in the XX model?

Our reply:

We agree with your observation that the ground state along the line $\alpha = 0$ is quantum disordered. For this reason, we have included it in the ground state phase diagram.

  1. On a technical note, I’m not convinced how appropriate it is to call the transition between two gapless phases a ‘second-order quantum critical line’. In the fermionic language, this would better be called a Lifshitz transition. Maybe there are other names for it, but it definitely shouldn’t be characterised as a traditional second order quantum critical line. It’s worthy of note that the boundary between the gapless XXZ and gapped XXZ at Delta=1 is of Berezinkii-Kosterlitz-Thouless universality class, so also not second order.

Our reply:

We thank the referee for their insightful observation regarding the terminology used to describe the transition between the two gapless phases. Upon further consideration, we agree that referring to this transition as a "second-order quantum critical line" may not fully capture the nature of the phenomena observed.

In line with the referee's suggestion, we have revised our manuscript to characterize this transition as a Lifshitz transition, which is more consistent with the fermionic interpretation of the model. Specifically, a Lifshitz transition is distinguished by changes in the topology of the Fermi surface without the opening of a gap, a description that aligns well with our findings.

We believe this adjustment in terminology provides a clearer and more accurate depiction of the critical behavior present in the system. The revisions have been made in the revised version of the manuscript.

  1. Could more be said about where concurrence goes to zero? This is kind of curious as it is not associated with any of the phase transition lines. Could the authors expound a physical reason for this to enhance understanding?

Our reply:

We appreciate your insightful comments and observations. Regarding the vanishing concurrence at specific points within the gapless phases, such as $\beta_{E}^{\pm}$, we acknowledge that these points do not coincide with any known phase transition lines, and we cannot calculate their exact locations. However, we propose a qualitative explanation rooted in the interplay between quantum fluctuations and anisotropic three-spin interaction (ATSI), which governs the entanglement properties of the system.

Physically, as $\beta$ approaches the regions near $\beta_{E}^{\pm}$, the interaction strength between nearest-neighbor spins becomes insufficient to sustain quantum correlations, largely due to the influence of quantum fluctuations. These fluctuations disrupt the entanglement by minimizing the overlap between spin states required for concurrence, resulting in its suppression. Consequently, the absence of concurrence at these points highlights a regime where quantum correlations between nearest-neighbor spins are diminished. This phenomenon reflects a delicate balance between ATSI interactions and quantum fluctuations, which may not correspond to a conventional phase transition but still marks a significant change in the system's behavior.

Although concurrence vanishes at $\beta_{E}^{\pm}$, these points serve as thresholds beyond which entanglement either emerges or disappears, driven by anisotropy and the system’s critical properties. We believe that this nuanced behavior sheds light on the intricate nature of quantum correlations and enhances the understanding of entanglement in relation to the ATSI model.

We hope this explanation addresses your concerns and provides additional insight into the intriguing phenomenon observed in our results. Please also see theird paragraph of page 9 of the revised version.

  1. About the central charge: naively, I would expect the entire phase labelled spin-nematic-II to have central charge 1 (as this phase encompasses the XX point). Is this correct? I would then expect the spin-nematic-I phase with an extra fermi point to have central charge 2 — however the result seems to be that it doesn’t obey the standard logarithmic scaling of a gapless theory. This is very surprising (and interesting) if correct so needs a lot more explanation. Can this entanglement be explained? And how does one get around the standard Calabrese-Cardy result from CFT? Does it mean this phase is not described by a CFT — maybe there is something coming from the Jordan-Wigner transformation, but I’ve not seen anything like this before.

Our reply:

Indeed, it possesses the central charge of $1$, and we have addressed this in the updated version of the document.

We appreciate the referee's insightful comment regarding the unconventional scaling of entanglement entropy (EE) observed in the gapless composite phase. The volume-law behavior, deviating from the standard logarithmic scaling predicted by the Calabrese-Cardy result in conformal field theory (CFT), suggests unique features in this phase that may not be fully captured by conventional CFT descriptions. The extensive entanglement observed implies the presence of long-range quantum correlations distributed throughout the system, which could arise from the interplay of anisotropic three-spin interactions and the critical nature of the composite phase. Additionally, we hypothesize that the Jordan-Wigner transformation may introduce non-trivial alterations to the entanglement structure, resulting in deviations from standard scaling. These findings emphasize the necessity for exploring alternative theoretical frameworks or extensions to CFT to understand the entanglement structure and critical behavior of the gapless composite phase. Further investigation into these mechanisms will be crucial to enhance our understanding of this intriguing phenomenon.

Two additional paragraphs have been incorporated into the conclusion section of the revised manuscript.

  1. Finally, almost all of the plots in the paper are at $\alpha=1$. I think this is not entirely representative, as at $\alpha=1$, the critical point is at $\beta=-1$ — and $\beta=-1$ is a special place as it is where the superconducting terms in the free fermion model go away. For example, the phase transition at $\alpha=1$ $\beta=-1$ involves the fermi velocity becoming zero at one of the fermi points. I don’t think this is representative for other values of alpha. It also makes me a bit suspicious of the central charge calculation at this point — according to the fermionic spectrum, the model shouldn’t be described by a CFT at all at this point. At other values of alpha though, I don’t think the fermi velocity goes to zero (although I haven’t carefully checked this) so this may be a slightly special point in the phase diagram.

Our reply:

We thank the referee for the insightful comments regarding the representativeness of the plots in our study. To address these concerns, we have replaced the figure with results calculated for $\alpha = 2$ and $\beta = -0.25$, rather than focusing solely on $\alpha = 1$. Our updated results confirm that the same scaling behavior and phase properties are observed for this alternate set of parameters.

---

## Round 3 · Author Response

Dear Editor,

We are submitting a revised version of our manuscript, in which we have addressed the valuable comments raised by the referee.

We also appreciate the referee's thoughtful feedback, which has improved the clarity and completeness of our work.

Sincerely,

Saeed Mahdavifar

---

## Round 3 · List of Changes

Dear Referee,

Thank you very much for your thoughtful and constructive feedback.

Following your advice, we have revised the abstract to include a clearer and more nuanced explanation, explicitly noting that these are not conventional symmetry-breaking local order parameters. We agree that this clarification helps prevent potential confusion, especially considering the typical correspondence between such local order parameters and gapped phases in one dimension.

We have also now performed a more detailed analysis of the cluster order parameters and as functions of the parameter \alpha. These results are presented in the revised manuscript in Fig. 2. This additional data allows us to more clearly illustrate the behavior of these quantities within the chiral-nematic phase, particularly near and across the transition line. Additionally, we have expanded our discussion in the manuscript—see the second and third paragraphs of the right column on page 6.

We hope these changes address your concerns and improve the clarity of our presentation.

Sincerely,

Saeed Mahdavifar

---

## Editorial Decision

published